# Control of Beany Flavor from Soybean Protein Raw Material in Plant-Based Meat Analog Processing

**DOI:** 10.3390/foods12050923

**Published:** 2023-02-22

**Authors:** Lingyu Yang, Tianyu Zhang, He Li, Tianpeng Chen, Xinqi Liu

**Affiliations:** 1National Soybean Processing Industry Technology Innovation Center, School of Food and Health, Beijing Technology and Business University (BTBU), Beijing 100048, China; 2Puluting (Hebei) Protein Biotechnology Research Limited Company, Handan 056000, China; 3Shandong Gulin Food Technology Limited Company, Yantai 264010, China

**Keywords:** soybean protein, drying method, storage conditions, extrusion processing, interaction, beany flavor

## Abstract

The development of plant-based meat analogs is currently hindered by the beany flavor generated by raw soybean protein and extrusion processing. Wide concern has led to extensive research on the generation and control of this unwanted flavor, as an understanding of its formation in raw protein and extrusion processing and methods through which to control its retention and release are of great significance for obtaining ideal flavor and maximizing food quality. This study examines the formation of beany flavor during extrusion processing as well as the influence of interaction between soybean protein and beany flavor compounds on the retention and release of the undesirable flavor. This paper discusses ways to maximize control over the formation of beany flavor during the drying and storage of raw materials and methods to reduce beany flavor in products by adjusting extrusion parameters. The degree of interaction between soybean protein and beany compounds was found to be dependent on conditions such as heat treatment and ultrasonic treatment. Finally, future research directions are proposed and prospected. This paper thus provides a reference for the control of beany flavor during the processing, storage, and extrusion of soybean raw materials used in the fast-growing plant-based meat analog industry.

## 1. Introduction

Plant-based meat analogs are attracting increasing scientific and consumer attention because of the role they play in effectively alleviating a future shortage of animal protein supply and reducing the negative impacts of environmental pollution and disease encountered in traditional animal husbandry and animal food processing. Moreover, plant-based alternatives can also offer high nutritional value, advantageously meeting the nutritional needs of special groups (such as bodybuilders). They have thus become a research hotspot with significant market prospects [1,2]. Plant-based meat analogs, also known as plant-based protein meats, are mainly made from textured protein through seasoning, cooking, or reconstitution reprocessing [3] and commonly manufactured using soybean protein, wheat protein, or other plant proteins as their main raw material. These are pretreated via crushing, mixing, heating, and direct steam, followed by mixing, extrusion, shearing, melting, shaping, and a number of other physical treatments carried out by an extruder. During this process, the plant protein is changed by physical or chemical means to arrange the molecules in an orderly and homogenous organizational structure, forming plant protein products that are fibrous and taste similar to animal muscle [4,5]. Soybean protein is widely used as the raw material for textured protein because it is abundant and contains eight kinds of essential amino acids required by the human body [6]. Plant-based meat analogs are currently considered to be the most promising meat substitutes; however, their pervasive flavors remain a challenge that has largely hindered their acceptance by customers [7].

Generally speaking, volatile components that can produce unpleasant sensations when generated by or released or retained from food raw materials and their processing are identified as off-flavors [8]. The off-flavor in plant-based meat analogs is commonly described as a beany flavor, which presents characteristics of fat, grass, and earth [9]. Beany flavor is generated mainly by aldehydes, ketones, alcohols, and other small molecular volatile compounds [10]. The main reason for the beany flavor in soybean protein is that the unsaturated fatty acids (linoleic and linolenic acids) rich in soybean are easily oxidized to form hydroperoxides under the action of lipoxygenase, which are then further degraded to form aldehydes, ketones, and alcohols, among others [11]. People’s perception of this beany flavor is largely dependent on its molecular composition, release, and retention. The structure and properties of food determine the stability of volatile flavor compounds in the food matrix [12]. Studies have proven that once a beany flavor has formed in food, it is difficult to remove. The food industry has made every effort to reduce the formation of beany flavor (such as through lipid oxidation); however, even bean products produced from soybean varieties without lipoxygenase still have that beany flavor [13]. In soybean protein raw materials and extrusion processing, the proteins provide complex binding sites that interact with beany flavor compounds. In addition, steric hindrance formed by structural changes in the finished product blocks enhances the release of beany flavor substances, which are evident to our olfactory sensors [14]. Therefore, the study of drying methods and storage conditions for soybean protein raw materials as well as the influential mechanisms of extrusion parameters and the environment on the beany flavor during extrusion processing can provide a theoretical basis for flavor improvement.

In this paper, both the formation and removal of beany flavor substances in soybean protein raw materials and textured proteins in the extrusion process are systematically discussed. Furthermore, recent research findings pertaining to the regulation of beany flavor from soybean protein are reviewed. The paper focuses on the effect of the extrusion process on beany flavor as well as the effect of protein binding to the beany flavor compounds. This also includes research progress on drying and storage methods for soybean protein raw materials. Finally, future research directions are proposed and predicted. Thus, this review serves as a valuable reference for the improvement of flavor characteristics in plant-based meat analog products.

## 2. Main Components and Production Mechanism of Beany Flavor in Soybean Protein

### 2.1. Main Beany Flavor Compounds in Soybean Protein

Research has shown that the beany flavor is an undesirable mixture of a variety of volatile small molecular compounds, including mainly “beany”, “green”, “bitter”, “earthy”, and other unpleasant flavors [15]. More than 30 kinds of volatile substances have thus far been related to the beany flavor: predominantly aldehydes, ketones, and alcohols [11]. Achouri et al. [16] found that hexanal, hexanol, 1-octen-3-ol, and 2-pentylfuran had the greatest impact on beany flavor. The odor threshold of beany flavor compounds is relatively low, and it is known that hexanal has a green flavor and that its threshold in water is very low (0.0045 ppb) [17].

### 2.2. Mechanism of Beany Flavor Generation in Soybean Protein Raw Material

The off-flavor substances to which the beany flavor in soybean protein can be attributed are produced mainly via the degradation of polyunsaturated fatty acid derivatives promoted by lipoxygenase (LOX). LOX in soybean exists mainly in the cotyledons near the soybean epidermis, which is removed during the refining of soybean seeds. However, even very low levels of LOX activity can cause lipid degradation problems during dry powder preparation and beating [18], as shown in Figure 1. During the processing of soybean protein, polyunsaturated fatty acids with a (Z, Z)-1,4-pentadiene structure can be specifically catalyzed by LOX to form hydroperoxides. These compounds are then further degraded to produce aldehydes, ketones, alcohols, and other volatile compounds, leading to the formation of the beany flavor [19]. Furthermore, the beany flavor in soybean protein raw materials is additionally affected by factors such as the automatic oxidation and photooxidation of lipids due to environmental factors, including high temperature, light, and the partial pressure of oxygen during drying and storage. Automatic oxidation is a free-radical reaction initiated by singlet oxygen produced by natural pigments. Highly active singlet oxygen attacks the hydrogen on the α-methylene to form alkyl radicals (R•), which further absorb oxygen from the air to produce peroxide radicals (ROO•), thus forming hydroperoxide and producing 2-pentylfuran, 1-octene-3-ol, and other beany-flavor-related compounds [20]. Unlike automatic oxidation, photooxidation is the direct action of singlet oxygen on the double bonds of unsaturated fatty acids to form hydrogen peroxide [21].

### 2.3. Mechanism of Beany Flavor Generation during Extrusion

Soybean protein raw materials are often extruded by extrusion technology to produce textured protein, which is then further processed into plant-based meat analogs [22]. Numerous reactions occur during the extrusion of food materials based on the material being extruded as well as the distinct extrusion conditions. The relationship between soybean protein and beany flavor substances in the extrusion process can be described as follows: during thermal processing, thermal degradation of protein occurs, in which the generated amino acids undergo deamination and decarboxylation to generate volatile alcohols, aldehydes, and sulfides, among others [23]. When heated, the mixture of sugar and amino acids undergoes the Maillard reaction and Strecker degradation to produce a series of volatile beany substances [24]. The amino acids generated during protein hydrolysis are decomposed into alpha-keto acids, amines, and carbon dioxide. The α-keto acids produced by the breakdown of amino acids are further converted into sugar and other soybean flavor precursors [25].

The formation of beany substances in the extrusion of sugar and soybean protein raw materials can be divided into three stages. In the first stage, the protein is degraded by heat to produce small peptides and free amino acids, and the hydrogen bond begins to break. The precursor of soybean flavor substances is partially pyrolyzed to produce Amadori compounds, and the volatile soybean flavor substances begin to decompose. During the second stage, the protein is heated, and the Amadori compounds react to produce a large number of carbonyl compounds and precursors of beany compounds, while some Amadori compounds begin to rearrange. During the third stage, the conformation of the extruded protein starts again, and the protein generates aldehydes, furans, and other compounds through the Strecker degradation reaction [26]. Glutamic acid may produce substances with a bitter almond flavor after the Maillard reaction [27]. When there is no sugar added during the extrusion process, and soybean protein is the main raw material, the Maillard reaction is weak. Moreover, some compounds produced by the Maillard reaction, such as pyrazine, can improve protein flavor [28]. As shown in Figure 1, the high-temperature environment of the extrusion process causes the partial lipid residue in the soybean raw material to be oxidized automatically, forming a beany flavor.

## 3. Retention and Release of Beany Flavor Compounds in Soybean Protein Raw Materials and Textured Proteins

### 3.1. Interaction Mechanism of Beany Flavor Compounds and Soybean Protein

Proteins in the food matrix have no flavor of their own; however, they can combine or absorb flavor compounds, thereby affecting flavor perception in the process of consumption [29]. On the one hand, due to the ability of protein molecules to combine and isolate lipophilic flavor molecules, an off-flavor can be combined into food, which significantly affects its sensory and consumption [30]. Some small molecular volatile compounds (such as aldehydes and ketones) can thus remain in food and assert their beany flavor through the interaction with protein. On the other hand, however, protein can also be used as a flavor carrier, providing and retaining desirable food flavor substances that can actually improve the flavor of products [31]. Therefore, a basic understanding of the interaction mechanism between beany flavor compounds and proteins and the factors affecting their binding can be used to predict and control the release/retention behavior of beany flavor compounds.

The protein–flavor interaction mechanism can be both reversible and irreversible [32], as shown in Figure 2. Reversible interactions are mainly the binding of volatile compounds with protein -COOH, -SH, -NH2, and -OH, such as van der Waals forces, ionic bond, hydrogen bond, and hydrophobic interactions [33]. The irreversible interactions are mainly due to the formation of strong covalent bonds between volatile compounds and protein -S-S-, -SH, and -NH2. Proteins with a high content of lysine, arginine, and cysteine are more likely to covalently bind with volatile compounds [34]. In most cases, the interaction between protein and beany flavor compounds is reversible, and this combination significantly affects the ideal flavor in food [35].

### 3.2. Interaction between Different Beany Flavor Compounds and Soybean Protein

#### 3.2.1. Aldehydes

Aldehydes are considered to be the main component of the beany flavor. As a group of compounds, they are characterized by a hydrogen atom (-H) on the carbonyl group (C=O). Their sensory threshold is low, and it is an aldehyde with “fresh, fat and green” sensory characteristics that is considered to be the most influential and most common beany flavor compound [36]. Among aldehydes, hexanal is the most typical, but other aldehydes that produce peculiar flavors in plants include heptaldehyde, nonanal, octanoic aldehyde, and (E, E)-2,4-dodecylaldehyde. These “fatty” aldehydes are formed by the degradation of polyunsaturated fatty acids through autoxidation (spontaneous reaction with oxygen), photooxidation (photocatalysis), and enzymatic catalysis [37]. Kim and Min [38] systematically summarized the potential chemical interaction between soybean protein and volatile aldehydes and pointed out that volatile aldehydes can condense with free amino acids or the free amino acids of the protein to form Schiff bases. These bases subsequently react with another hexanal molecule to form aldehyde alcohol, which then hydrolyzes to form an aldehyde product. Aldehydes sometimes form either reversible hydrophobic interactions or irreversible covalent bonds with the amino and sulfhydryl groups of soybean protein.

#### 3.2.2. Alcohols

Alcohol is one of the main products of the oxidation increase of linoleic acid during storage. Beany flavor compound 1-octene-3-ol is a kind of vinyl alcohol with a strong mushroom flavor [39]. Interactions between alcohol and soybean protein are affected by the chain length and temperature of the compound. Kim and Maga [40] studied the retention of a series of volatile compounds with different chain lengths (C6, C8, and C10) and functional groups (acids, alcohols, and aldehydes) during high-temperature extrusion. The highest total retention rate was with alcohol, and the lowest was with the aldehydes. It was found that the amount of bound volatiles increased as the chain length of alcohol volatile compounds increased, and fewer bound volatiles were found at higher extrusion temperatures (115–135 °C).

#### 3.2.3. Ketones

Ketones are also flavor compounds produced by lipid degradation. In their structure, ketones have the same functional carbonyl group (C=O) as aldehydes and alcohols but with no hydrogen atom. There is only a reversible weak hydrophobic interaction between ketones and proteins. The steric hindrance produced by the 2-position ketone group in the ketone may limit hydrophobic interaction and generally prevent the binding with protein [41]. Regardless of the protein source or separation method, the retention of aldehydes and ketones increases significantly with the increase in carbon number, indicating that these interactions are mainly hydrophobic in nature. Compared with aldehydes, however, ketones have lower hydrophobicity and lower binding ability with proteins [42].

### 3.3. Release and Retention of Beany Flavor during Extrusion

The extrusion process can inhibit the formation of some beany compounds while increasing the release of others. As shown in Figure 1, extrusion technology can effectively inactivate lipoxygenase and prevent lipid oxidation induced by lipoxygenase through its high-temperature and -pressure environment, thereby improving the flavor of textured protein. The principle underlying this phenomenon may be that high temperature can change the structure of lipoxygenase and reduce the efficiency of lipid oxidation [43]. Samard et al. [22] found that the beany flavor of soybean protein can be effectively reduced in the extrusion process and that increasing the extrusion temperature can also greatly inhibit the production of beany flavor in the preparation and processing of soybean protein. Due to the high-temperature and -pressure conditions in the extruder barrel, an ambient pressure difference is generated at the end of the extruder, and the steam (at atmospheric pressure) is rapidly ejected. The volatile beany flavors are volatilized into the air with the flashing of water, thus releasing the contributing flavor substances [44].

As described above, some beany compounds are released during the process of extrusion, while others remain. There are two main factors involved in the retention of volatile beany flavor compounds in extruded textured proteins: First, the blocking or interception effect is caused by macromolecules and structural organization [45]. As shown in Figure 1, in the generation of textured protein through extrusion, when the soybean protein raw material is expanded by the high-temperature, high-pressure, and high-shear environment towards the outlet of the extrusion die, the textured protein forms a gel network structure [46], which acts as a steric barrier that contains, wraps, and retains the beany compounds. In addition, these may also be wrapped within the solid matrix due to the sudden increase in the viscosity of the extrusion [47].

Second, during the extrusion process, the soybean protein denatures under the high-temperature, high-pressure, and high-shear environment, and its secondary bonds change [48], thus altering the protein structure. Aldehydes and alcohols are more likely to have non-covalent cross-linking with soybean protein, resulting in the retention of beany flavor components and affecting the flavor characteristics of the extrusion. Yuan et al. [49] found that the retention of total hexanal and total volatile flavor compounds was directly proportional to the content of soybean protein. When the protein content is high, increasing gel network structures are formed inside the extruded textured protein, which may increase the number of attachment sites for volatile beany substances, leading to an increase in beany substance content. The relative volatility of beany flavor compounds is strongly affected by their reversible interaction with the protein components in the food [33]. The greater the interaction, the less the relative volatility, and the more the beany flavor remains during the steam-release processes (such as at the outlet of the extruder mold).

## 4. Control of Beany Flavor in Soybean Protein Raw Materials

### 4.1. Effect of Different Treatment Conditions on Beany Flavor

The elimination and control of the beany flavor of soybean protein raw materials can be achieved mainly by preventing the oxidation and reducing the activity of LOX. Soybean protein is usually made into powder for more convenient use as a food raw material. During the milling process, the soybean flavor can be effectively reduced via the removal of the lipoxygenase in the cotyledons close to the soybean skin (the peeling rate should be more than 90%) during pretreatment [50]. Many methods have been applied to inhibit LOX activity in soybean protein raw materials. Heating is the most commonly applied method to remove the beany flavor during soybean processing since heat treatment can effectively reduce the activity of LOX, thereby reducing the production of beany compounds [51]. Moreover, volatile compounds produced during heat treatment, such as pyrazine and alkylpyrazine, also contribute a certain masking effect on the beany flavor [52]. At present, in addition to traditional heating methods, microwave heating and radio frequency heating are also widely used to remove the beany flavor of soybean products. Radiation, high-frequency electromagnetic fields, and hormone treatment can also reduce the activity of fat synthetase in soybean [53,54]. In addition, the activity of LOX can be reduced by adjusting the pH, thus removing the beany flavor in soybean. In an alkaline medium (pH = 8.3), the activity of LOX may be inhibited [55]. It was reported that pressure cooking (120 °C, 2 min) was used as a pretreatment, followed by 0.25% (*w*/*v*) sodium bicarbonate solution to soak the soybeans at room temperature. The obtained soy protein drink was less beany and tasted better than that obtained using the normal soaking method [54]. This may be attributed to the addition of sodium bicarbonate, which impacts the pH condition required for optimal LOX activation.

### 4.2. Effect of Different Drying Methods on Beany Flavor

Food drying has been widely used in the modern food industry as an important processing operation. Its purpose is to reduce moisture content in the raw material and effectively prevent the breeding of microorganisms in the material, the occurrence of chemical reactions with water, and the deterioration of the soybean protein, thus extending shelf life and reducing freight costs while at the same time facilitating the subsequent deep processing [56]. The drying methods used in the production of soybean protein powder greatly affect the flavor characteristics of the final product. Common food drying methods include hot air drying, microwave drying, freeze drying, and spray drying [57]. The content of volatile off-flavor substances produced by these drying methods differ, and each also has its own advantages and disadvantages, as summarized in Table 1.

The most traditional soybean drying method is solar drying; however, this is a slow process and is affected by unstable weather conditions. Compared with non-dried soybeans, sun-dried soybeans generally have low porosity and high apparent density [58]. Protein powder from soybeans that are first dried in the sun and then baked in the oven has a low beany flavor and a sweet taste. Drying in an oven at 50 °C for 3 h is traditionally the preferred method to produce soybean protein powder with good flavor and a sweet taste. Oven drying can improve sensory characteristics by reducing the beany flavor of soybean protein while increasing its sweet taste and baked flavor [51,59]. Drying soybeans in a drum roaster leads to a significant decrease in their moisture and free amino acid content. Compared with raw soybean, there is no significant difference in total lipids and fatty acids of dry roast. The oil absorption of oil-roasted soybean in the process of oil roasting led to a significantly higher content of total lipids and individual fatty acids than that of dry-roasted soybean and raw soybean [60]. In addition to these basic sunlight- and oven-drying methods, the most commonly industrial methods are spray drying and freeze drying. Freeze drying removes water from frozen products by sublimation. The powder obtained after freeze drying usually has low bulk density, high porosity, and good flavor and taste [61,62]. However, the freeze drying process is time-consuming and requires a great deal of energy. As an alternative, one study [63] showed that solid dispersion spray drying can also reduce the beany flavor in soy protein isolate. This reduction in flavor can be explained by the loss of volatile compounds during water evaporation and the subsequent changes in the tertiary structure of the soybean protein and soybean protein-polysaccharide complex, which reduce the combination of volatile compounds and proteins. In addition, microwave vacuum-drying technology has the advantages of rapid heating speed, high efficiency, good controllability, and good sanitation. Compared with microwave, vacuum, and freeze drying, however, solar and oven drying require little energy consumption, which makes them inexpensive and suitable for agricultural environments with limited resources [64].

**Table 1 foods-12-00923-t001:** Advantages and disadvantages of different protein drying methods.

Drying Method	Advantages	Disadvantages	Reference
Solar drying	Low energy consumption; inexpensive	Long process; poor flavor; drying speed is difficult to control; product cannot be dried evenly; slow drying speed; exposure to environmental pollution; requires large space for drying.	[58]
Oven drying	Low energy consumption; inexpensive; good flavor	Small processing capacity	[64]
Freeze drying	Minimal degradation; generally high retention rate and efficient extraction of nutrients from raw products	Expensive; slow drying speed; low product output	[61,62]
Spray drying	Rapid drying time; continuous operation; full automated control	Large equipment; low thermal efficiency; high requirements with two-stage separation equipment	[65]
Low-temperature drying	Inexpensive; eliminates need for direct sunlight	Slow speed of drying; loss of nutrients in protein	[66]
Hot air drying	Low equipment investment cost; low operation cost; high thermal efficiency; avoids heat loss	Covers a large area	[67]
Hot-air-assisted radio frequency drying	Strong penetration; stable temperature control; drying uniformity	Poor economic benefits	[68]
Microwave drying	Short starting time; reduced drying time; improved extraction of bioactive compounds	Bumping, degradation of heat-sensitive components	[69]
Convective drying	High processing economy; high drying rate; energy efficient; not affected by weather	Small drying system; low heat and mass transfer efficiency; deteriorated product quality; bioactive compounds degraded	[56]
Microwave vacuum drying	Maintains or accelerates microwave drying; eliminates need for high temperature; increases drying speed; reduces total drying time; minimizes oxidation reaction	High cost; high moisture evaporation; low capacity of the vacuum pump	[69]
Swelling drying	Improved volatile and non-volatile molecules extraction; strengthened drying dynamics; reduced energy consumption	Large amount of steam; expansion and cracking of plant materials; structure of materials change	[70]

### 4.3. Effects of Different Storage Conditions on Beany Flavor

Generally, dried soybean protein powder is stored after packaging; however, storage temperature, humidity, and light conditions are key factors affecting the production of beany flavor substances. While a large number of lipids (triglycerides) are removed during the milling of soybean protein, some residues usually remain, and these are easily oxidized and degraded in subsequent storage and extrusion, resulting in the production of some volatile beany flavor compounds. Changes in the packing and storage conditions of raw materials are therefore required to protect soybean protein products from lipid oxidation, reduce their light exposure, and thus reduce the beany flavor generated by the oxidation of singlet oxygen.

#### 4.3.1. Temperature Conditions

Storage temperature is a key factor affecting the beany flavor of protein. Da Silva Pinto et al. [71] studied the effect of different storage temperatures on the antioxidant activity of defatted soybean protein powder and soybean protein isolate (SPI) over one year. The study showed that the antioxidant activity of the samples stored at 42 °C decreased by 14–40% compared with those stored at −18 °C. Higher-temperature conditions lead to more drastic changes during storage, which are consistent with lipid oxidation in food. Therefore, soybean protein should be maintained at a low temperature to reduce the production of beany flavor components during storage.

#### 4.3.2. Humidity Conditions

Water activity (Aw) is a basic parameter of food stability. The safe moisture value of soybean protein powder affects its shelf life. One study found that, from the perspective of microbial stability, an Aw value of below 0.7 was conducive to long-term shelf life at room temperature. During long-term storage, however, relative humidity caused the quality of soybean to decline. No enzyme activity was observed in water activity below 0.3, and the reaction rate was generally very low in water activity below 0.6 [71]. Soybean protein components may be particularly susceptible to oxidation, especially in the form of hydration. At low water content and water activity, lipid oxidation will actually be very low. When the effects of Aw (0.11–0.75) on the reactivity of benzaldehyde, allyl thiocyanate, citral, and dimethyl sulfide with soybean protein were studied, the results showed that lower Aw resulted in a reduced reaction [13].

#### 4.3.3. Lighting Conditions

Under light, soybean protein powder is susceptible to singlet oxygen oxidation, which also causes its beany flavor. In addition, light can promote the automatic oxidation reaction of unsaturated fatty acids. Lee et al. [72] compared the volatile compounds in soybean protein powder under both light and dark conditions at 30 °C by combining solid-phase microextraction with gas chromatography–mass spectrometry and sensory evaluation. From days 0 to 6, the content of volatile flavor substances in the soybean protein powder increased by 60% and 300% with the extension of storage time, which clearly showed the important role played by light in the formation of volatile compounds during the storage of soybean protein powder. In addition, 2-pentylfuran, 1-octene, and 2-heptylenaldehyde were only present in the soybean protein powder stored under light, and the sensory odor score of the samples stored under light was significantly higher than that of the samples stored in darkness. These results demonstrated that the preservation of soybean protein raw materials in dark rather than light conditions is an effective way to improve their flavor.

#### 4.3.4. Inflatable Components

The production of beany flavor can be reduced mainly by isolating the product from oxygen and light. The formation of beany flavor compounds is affected by the air components in stored food, while a hypoxic environment reduces lipid oxidation; exposure of packaging to light may cause an increase in oxygen partial pressure, while the removal of oxygen is conducive to the inhibition of aldehyde formation during storage. Ozone is an effective antibacterial agent that could inhibit the growth of microorganisms and the release of enzymes in the ethanol fermentation pathway. Compared with vacuum packaging, nitrogen treatment has also shown effectivity in the inhibition of lipid oxidation, which can delay the formation of beany flavor compounds [73].

### 4.4. Effects of Different Extraction Methods on Beany Flavor

In order to obtain soybean protein isolate without an off-flavor, the complete removal of phospholipids and free fatty acids is necessary. Ideally, the precursor and the compounds that cause the flavor should be removed with only minimal changes to the functional characteristics of SPI. However, a certain amount of residual lipid contents tends to remain in soybean protein products, which varies according to the extraction method [74]. The various extraction methods used to reduce flavor compounds in soybean protein products are discussed as follows:

Salt-out method: This method is employed if the phospholipids in the SPI are mainly linked to oleic acid and some other secondary protein components. Using this principle, Samoto et al. [75] studied the use of ammonium sulfate fractionation to remove phospholipid-binding proteins. Phospholipid-associated protein was salted to 30%~40% in a saturated ammonium sulfate solution. Through thin-layer chromatography analysis, this fraction was found to contain a large number of neutral lipids, phospholipids, and glycolipids. One of the disadvantages of this method is that it is not suitable for large-scale operations.

Solvent extraction method: Polar lipids are extracted with solvents such as methanol, ethanol, isopropanol, hexane/alcohol azeotrope, and water acetone in an alcohol–water solution (78% to 97% *v*/*v*). These treatments have been proven to be very effective in reducing the phospholipid level in SPI while simultaneously improving its flavor. However, exposure of SPI to organic solvents can lead to protein denaturation and the loss of solubility as well as other functional characteristics that are critical to the function of protein in food [76].

Alkali solution and acid precipitation method: The precipitate formed by this method is found to be a lipid-related protein, the yield of which is approximately 31% of the total protein in the original protein extract, which is similar to the yield obtained by ammonium sulfate fractionation. The disadvantage of this method is that it leads to a low yield of lipoprotein-free SPI (~70%). In addition, the residual fat content and the degree of flavor generation in the final SPI products have not yet been reported. These problems require research investigation [75].

## 5. Effects of Raw Material Characteristics and Environmental Factors on Beany Flavor Retention

Minimal adsorption of the beany flavor compounds by protein and their maximum release into the air will reduce the beany flavor in raw materials and their subsequent food products, thereby improving consumer acceptance. Factors affecting the interaction between food protein and flavor include the chemical properties and reactivity of the protein and flavor compounds, the concentration and structure of flavor compounds [12], and the number and structure of protein side chains as well as environmental factors such as pH, temperature, ionic strength conditions, pressure, and oxidation conditions [77]. Table 2 summarizes the effects of raw materials, flavor characteristics, and different treatment methods on the interaction between protein and beany flavor compounds.

Proteins provide complex chemical sites that interact with flavor compounds. When beany flavor compounds combine with proteins through hydrophobic interaction, the factors that change the protein conformation and expose or destroy the internal hydrophobic region also change the binding affinity of beany flavor compounds [78]. Therefore, targeted modification of the soybean protein structure or the adjustment of key external factors in the food system can effectively change the binding affinity between protein and beany flavor compounds, thus eliminating the beany flavor. Active functional groups such as hydroxyl, sulfhydryl, and carbonyl can affect both the chemical reactivity of beany flavor compounds and the physical or chemical interaction (adsorption, oxidation, and polymerization) between beany flavor and protein [79]. In food systems, when flavor molecules are combined, the conformation of protein may change. In general, expanded or moderately denatured proteins can provide a large number of binding sites to attract flavor ligands with active groups [14], so any factor affecting the structure of protein can affect its flavor binding ability. Thus, the modification of soybean protein via chemical or enzymatic methods can change its binding affinity with beany flavor compounds. The addition of malondialdehyde (MDA) into a soybean protein solution reportedly changed the binding ability between the soybean protein and aldehydes. When the concentration of MDA exceeded 2.5 mM, the hydrophobic interaction between the SPI and compounds such as hexanal and nonanal was significantly reduced [80]. In another study, the binding affinity of vanillin and maltol with soybean protein decreased at 25 °C after deamination of the protein glutaminase (PG) via equilibrium dialysis (ultrafiltration) technology [81]. The molecular structure of volatile beany flavor substances also affects the combination of protein and beany flavor substances. For example, the affinity of butanol to protein is higher than that of hexanol [35]. At present, few studies have researched the factors that affect the interactions between protein and beany flavor compounds for the removal of beany flavor. The development of new methods or technologies that can affect the combination of flavor compounds and protein is essential to maintain the sensory quality of food.

**Table 2 foods-12-00923-t002:** Effects of different factors on the interactions between protein and beany flavor compounds.

Influence Factor	Protein Type	Processing Method	Result	Reference
Flavor compound concentration	Soybean protein isolate	1% SPI, 0.04–0.16 mM flavors, propylene glycol dissolution, shaken for 24 h at 37 °C to reach equilibration.	Higher concentration of flavor compounds affected the hydrophobic sites of protein to a greater extent	[12]
Malondialdehyde	Soybean protein isolate	MDA and 40 mg/mL SPI shaking reaction for 24 h	Concentration of MDA was ≤1 mM, the combination of flavor and SPI ↑; MDA content ≥2.5 mM, the combination of hexanal and nonanal with SPI ↓	[80]
Preheat treatment	Soybean protein isolate	Preheating at 80 °C, 90 °C and 100 °C	The temperature of SPI aqueous solution ↑; the binding ability of SPI to HxAc and HpAc ↓	[82]
Disodium inosine	Soybean protein isolate	Addition of 25 mg/100 g inosine disodium to 220 g soy protein isolate	Isovaleraldehyde and ethyl butyrate retention rates were significantly ↑	[83]
Glutaminase deamination	Soybean protein isolate	Glutaminase deamidation reaction for 2 h	Binding effect of vanillin and maltol with SPI was stronger than that with DSPI	[81]
β-Cyclodextrin	Soybean protein	Injected addition of 20 mM β-CD stock solution to buffer solution at 25 °C for 24 h to reach equilibrium	SP-bound 2-nonanone ↓ in a concentration-dependent manner	[84]
Temperature	Soybean protein	Addition of 1 mL of 30 mM Tris-HCl buffer solution to 0.5 mL of 20 mg/mL protein solution	Structural changes; at 20 °C and 30 °C, the number of binding sites of glycine was higher ↑ than that of *p*-homologous glycine	[85]
pH	Soybean protein	In 0.05N HCl, 2% soybean protein solution heated at 95 °C for 30 min	SP was preferentially deamidated, but the peptide bond did not break obviously; improved protein flavor	[86]
High-temperature pretreatment and enzymatic hydrolysis	Soybean protein isolate	Pretreatment at 121 °C, under optimal enzyme conditions (pH 6.0 and 45 °C), and hydrolysis with 100 mg Protemex	Some volatile compounds such as hexanol, hexanal, and pentanol in SPI ↓	[87]
NaCl	Defatted soybean extract	Defatted soybean extract precipitated at pH 4.5	The defatted soybean extract combined flavor volatiles and non-protein substances	[88]
Na_2_SO_4_ and CaCl_2_	Defatted soybean extract	At pH 2.8, 30 mM Na_2_SO_4_, and 30 mM CaCl_2_, conventional centrifugation (10,000× *g* for 10 min)	SPI prepared from defatted soybean extract contained volatile flavor compounds ↓	[89]

SPI, soybean protein isolate; MDA, malondialdehyde; HxAc, hexyl acetate; HpAc, heptyl acetate; β-CD, β-cyclodextrin; SP, soybean protein. The arrow “↑”represents increased and “↓” represents decreased.

## 6. Control of Beany Flavor in Soybean Protein Extrusion Processing

### 6.1. Effects of Different Extrusion Parameters on Beany Flavor

Beany flavor can also be influenced during extrusion by manipulating the parameters of the extrusion process. The extrusion method is advantageously used to produce products with good storage quality, oxidation, and flavor stability characteristics [90]; however, the process subjects material to high-temperature and -pressure conditions, which promote the oxidation of lipids and proteins [91]. In high-temperature short-time (HTST) food extrusion, changes in the characteristics of raw materials, such as moisture content, and in extruder operating conditions, such as temperature and screw speed, can affect the sensory acceptance of extrudates [31]. Thus, the quality of final products is determined by the combination of process parameters and equipment design [92].

#### 6.1.1. Moisture Content

Water is an important medium for the formation of beany flavor substances. During the extrusion mixing stage, raw textured protein materials of both low and high moisture content undergo complex rearrangement under the high-temperature and high-shear conditions [93]. By optimizing the moisture content of such raw materials, more and richer flavor substances can be produced [94]. Products made via low-moisture extrusion (moisture content 10–35%) have been shown to present characteristics of drying and slight expansion, which are conducive to the release of beany flavor, while high-moisture extrusion (moisture content 40–80%) can produce products with a high moisture content that do not require replenishment with water and whose fiber structure is closer to real meat; however, this method tends to increase the content of beany flavor substances [31].

During low-moisture extrusion of textured protein, the swelling structure formed causes volatile beany flavor substances to volatilize with the evaporation of water, resulting in their reduced retention [95], which affects consumer perception of the final product. In low-water extrusion, the lower water content can result in reduced intensity of the beany flavor and better sensory acceptance of the extrusion. For example, compared with 40% water extrusion, 30% water extrusion of SPI was found to have a lower overall flavor intensity and provide a higher preference score for products [83]. The textured protein or plant-based meat analogs prepared by high-water extrusion exhibit more compact structures (indicating a lower degree of swelling), which is accompanied by changes in protein conformation [31]. These changes in protein structure may form space resistance or affect its binding ability with beany flavor compounds, thus inhibiting the release of beany flavor and retaining it. However, some studies have shown that the retention rate of volatile beany flavor compounds or added flavor enhancers decreased with the increase in water content during high-water extrusion. For example, it was found that in meat substitutes made by the high-moisture extrusion method, when the moisture content increased from 40% to 80%, the retention rate of total volatile flavor compounds and individual volatile flavor compounds decreased, dropping most sharply when the moisture content was greater than 60%. This phenomenon may have been due to the increase in water evaporation and the large release of volatiles [96].

#### 6.1.2. Extrusion Temperature

Extrusion temperature is another key factor in the production and retention of beany flavor since it has a significant effect on lipid oxidation. High-temperature inactivated lipoxygenase can inhibit fat oxidation and reduce the content of off-flavor substances. Zhu et al. [97] examined the effect of the soybean extrusion process on the inactivation of lipoxygenase-1, -2, and -3 under different temperature conditions. The results showed that with the increase of extrusion temperature, activity levels of the three lipoxygenases decreased significantly, with the resistance of lipoxygenase-2 enzyme to inactivation during extrusion the strongest. Thus, it was shown that extrusion temperatures above 120 °C could effectively inactivate lipoxygenase, thereby preventing adverse flavor formation reactions.

In addition, when extrusion temperature is raised to a certain extent, soybean protein is heated to form a melting state, which expands the structure of the textured protein generated by high shearing action, thereby more easily exposing the hydrophobic group of the protein. Protein’s strong adsorption capacity for some volatile flavor substances such as aldehydes is less affected by water evaporation during extrusion, resulting in a large retention rate. As extrusion temperature continues to rise, water evaporation in the textured protein increases, and additional beany flavor substances flow out, followed by the release of still more volatile beany flavor substances, resulting in a lower total retention rate. The soybean protein extruded at a melting temperature of approximately 150 °C is conducive to the production of meat analogs with a higher degree of texture, lighter color, and better sensory properties [48,98].

#### 6.1.3. Screw Speed

The effect of screw speed on volatile beany flavor substances may also be related to the structural changes to protein during extrusion. To a certain extent, an increase in screw speed during extrusion will lead to a higher probability of collision between protein and beany flavor substances, resulting in the enhanced adsorption capacity of protein to some volatile beany flavor substances and a higher retention rate in the extruded textured protein. Research has shown that an increase in the screw speed of the extruder and instantaneous cooling leads to a significant increase in matrix viscosity. In turn, this increase reduces the diffusivity of volatiles and fixes the beany flavor substances, while the volatiles are retained in the extrudates due to effective encapsulation in their solid matrix [99].

With continuous increase in screw speed, the retention of material in the barrel is shortened, which consequently reduces protein adsorption time and results in the release of more beany flavor substances. Furthermore, some beany flavor compounds are lost in water evaporation during the material extrusion process, further reducing the retention rate, and the raw material becomes broken or aggregated due to the shear stress [100].

### 6.2. Effects of Different Additives on Beany Flavor in Textured Protein

The addition of flavor-improvement substances to the raw material is a convenient method for the removal of beany flavor during soybean protein extrusion. Since the main components of beany flavor are small molecular aldehydes and alcohols, the addition of specific aldehyde and alcohol dehydrogenase to soybean products can play a certain role in mitigating the unwanted flavor. Soybean protein with no obvious beany flavor was reportedly obtained by treating the material with aldehyde dehydrogenase (ALDH) at 45 °C for 2.5 h [101]. The additions of other active ingredients to reduce the beany flavor have also been studied. Treatment with cellulase and pectinase were found to significantly reduce the beany flavor of the product while also significantly improving its sensory quality. Adding malt and flavonoids has also been shown to reduce beany flavor, mainly because wheat germ powder is rich in ALDH, which can catalyze the reaction of the beany flavor substances, while flavonoids can closely combine with lipoxygenase to reduce its enzymatic activity. Flavonoids such as quercetin, flagellin, and catechin can also be closely combined with the secondary structure of lipoxygenase to effectively reduce its catalytic efficiency [102]. In addition, allyl isothiocyanate can be used to improve flavor by reacting with protein, cracking disulfide bonds, or attacking free amino acid residues of arginine and lysine to form polymers, while vanillin interacts with amino acids or proteins through covalent bonds formed by Schiff base arrangement. In general, proteins containing higher levels of cysteine, lysine, and arginine residues may form covalent bonds, thus showing higher flavor-binding ability.

The presence of a third competitive compound in the food system has been reported to effectively reduce the binding rate of soybean protein and flavor compounds. As has been ascertained, reducing the adsorption of beany flavor and increasing the adsorption of aromatic compounds is an ideal method for flavor improvement [32]. Microbial transglutaminase (MTGase), MDA, and cyclodextrin (CD) are commonly used structural modification agents of soybean protein, which can interfere with the binding mechanism of protein–soybean flavor compounds and are expected to remove the beany flavor [80]. Arora and Damodaran [84] found that β-cyclodextrin could effectively inhibit the binding of soybean protein with 2-nonanone. The higher the concentration of β-cyclodextrin, the better the inhibition effect on beany flavor. When the concentration of β-cyclodextrin was 6 mM, the binding rate of soybean protein with 2-nonanone was significantly reduced by 94%. The simultaneous treatment of soybean protein with phospholipase A_2_ and cyclodextrin removed more than 92% of the flavor precursor in SPI [103]. Although the application of cyclodextrin has been shown to effectively remove beany flavor from the soybean formula, research on this phenomenon is still limited. In particular, it is not clear whether the cyclodextrin that traps the flavor compounds is removed or remains in the soybean product.

The addition of gluten powder in the extrusion process is known to be conducive to the formation of a fiber structure similar to that of meat. Furthermore, to a certain extent, increasing the content of gluten powder can improve the viscosity of raw materials in the barrel, form the fiber microstructure in textured protein and the large fiber structure interconnected with much smaller fibers [3], enhance the adsorption of protein on volatile beany flavor substances, and increase the retention rate. When the gluten powder continues to be increased, slight swelling occurs in the plant-based meat analogs at the moment of extrusion, which causes the surface of the extrudate to crack. Some volatile beany flavor substances are subsequently volatilized and released through the broken surface, resulting in a reduction in their retention rate [31]. Guo et al. [31], who studied the mixing and extrusion of wheat protein and soybean protein, found that the retention rate of volatile flavor substances in the product increased with the increase of wheat gluten content, indicating that the proper addition of plant protein with richer amino acid content can improve the adsorption capacity of plant protein meat and flavor substances.

## 7. Conclusions and Prospective Outlook

Despite extensive global studies of the formation and removal of beany flavor, no method has yet proven completely effective. The search for methods that can effectively eliminate or disguise the volatile off-flavor substances in textured protein is essential for the promotion of plant-based meat analogs. The beany flavor is generated from the peeling and milling of soybean raw materials and affected by the storage and production process. Therefore, in order to reduce the impact of beany flavor on textured protein, the drying and storage of raw materials as well as the production conditions should be controlled. The production of plant-based meat analogs is not simply a matter of copying the texture of meat in an attempt to imitate animal meat products. With the increasing application of soybean protein in food, it is necessary to deepen the understanding of the generation, retention, and release of beany flavor in the production process (mainly extrusion technology) of plant-based meat analogs. In the future, more attention should be paid to the mechanism of interactions between beany flavor compounds and proteins since the factors that affect this binding phenomenon may be helpful in predicting and controlling their release and retention behavior. Research into new methods and technologies that can eliminate the beany flavor should be strengthened and the combination of multiple methods considered. Moreover, the nutritional and safety problems known to exist in soybean protein products, such as flatulence factors, allergens, anti-nutrient substances, and insufficient trace elements, should be thoroughly investigated to overcome the challenges surrounding these factors.

## Figures and Tables

**Figure 1 foods-12-00923-f001:**
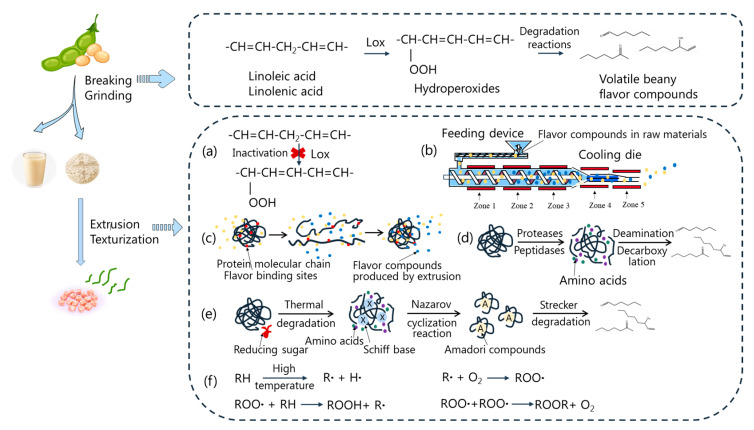
The formation, retention, and release of beany substances in soybean protein raw materials during processing: (**a**) high-temperature inactivation of lipoxygenase; (**b**) water evaporation releases beany compounds; (**c**) binding effect of protein on beany flavor compounds; (**d**) protein degradation produces beany flavor compounds; (**e**) Maillard reaction; (**f**) automatic oxidation of some unsaturated fatty acids.

**Figure 2 foods-12-00923-f002:**
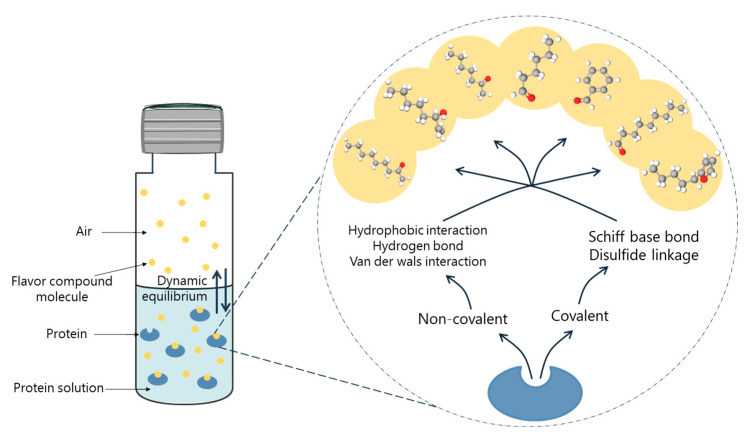
Flavor balance and interaction between flavor compounds and protein in headspace solid phase microextraction.

## Data Availability

No data were provided in the study.

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
