# Peer review of "Control of Beany Flavor from Soybean Protein Raw Material in Plant-Based Meat Analog Processing"

_foods, 2023, doi:10.3390/foods12050923_

Round 1

Reviewer 1 Report

"Control of beany flavor from soybean protein raw material in plant-based meat analog processing" is a review paper covering the area of which flavor components are present in soybean protein and how process parameters focused on meat analog processing affect them.

The paper is well-written and includes relevant sub-areas.

Comments:

The authors are referring too much to other review papers instead of the original papers. 

Line 37: include space before the reference.

Line 291-294: According to the reference - no significant differences were found for the dry-roasted in total lipids and fatty acid as compared to the raw. In addition, the study referred to did not use oven roasting - it used drum roasting. And in turns of amino acid content - the changes were in free amino acids. 

Line 543: The reference is a review paper - include the original study as a reference for the statement.

Line 544: 2-nonanone

Line 544-546: reduced as compared to what concentration?

Line: 546: 2-nonanone

Line 554-557: add reference - is this due to adsorption or just dilution of soy protein?

Line 557-559: add reference - Is this swelling occurring with the addition of pure gluten protein - or added starch into the system? What do you mean by crack? The fiber formation structure?

Line 559-561: reference number 3 is not looking into what the author is referring to. 

Reviewer 2 Report

Please rewrite the tables as 

table 1: make two separate columns for advantages and disadvantage

table 2: insert symbols for showing increase or decrease, and make the content shorter as provided text is too much and not easy to read instantly  

Reviewer 3 Report

The manuscripts aim ti review the consitions, process and compunds that are responsibles for beany off-flavours in soybean manufacturing and processing. It is an interesting paper. Information is concise and clear, easy to read and follow. Even it could be no very new, it is very useful. I have no suggestions to made nor additional comments.
